# Effectiveness of a home-based exercise program for improving upper limb function in community-dwelling older people: A pragmatic randomized controlled trial

Amanda Bates[1,2]*, Cathie Sherrington[2,3], Susan Furber[4,5], Heidi Gilchrist[2,3], Paul van den Dolder[6], Karen Ginn[7], Adrian Bauman[2], Michelle Kershaw[1], Lisa Franco[8], Cathy Chittenden[9], Anne Tiedemann[2,3]

**1** Health Promotion Service, Illawarra Shoalhaven Local Health District, Wollongong, Australia, **2** School of Public Health, Faculty of Medicine and Health, The University of Sydney, Sydney, Australia, **3** Institute for Musculoskeletal Health, The University of Sydney and Sydney Local Health District, Sydney, Australia, **4** Formerly Illawarra Shoalhaven Local Health District, Wollongong, Australia, **5** Formerly School of Health and Society, Faculty of the Arts, Social Sciences and Humanities, University of Wollongong, Wollongong, Australia, **6** Primary Care and Community Health, Nepean Blue Mountains Local Health District, Australia, **7** Faculty of Medicine and Health, The University of Sydney, Sydney, Australia, **8** Integrated Care and Priority Populations, Illawarra Shoalhaven Local Health District, Wollongong, Australia, **9** Department of Physiotherapy, Illawarra Shoalhaven Local Health District, Australia

* amanda.bates@health.nsw.gov.au

## Abstract

### Background

Upper limb dysfunction, including shoulder pain, is a major health issue for older people. Exercise is commonly used to treat shoulder disorders. This study aimed to determine the effect of a home-based upper limb exercise program on upper limb function compared to a lower limb exercise program, among community-dwelling people aged 65 years+.

### Methods

A randomized controlled trial was conducted. One group received a home-based exercise program targeting upper limb (UL) strength, mobility and function. The other group received a home-based exercise program targeting lower limb (LL) balance and strength. Both exercise programs were taught at three group-based sessions in weeks 1, 4 and 12 post-randomization. Participants were requested to complete exercises three times per week for 12 months. The primary outcome was UL function, measured with the self-report Disabilities of the Arm, Shoulder, and Hand (DASH) questionnaire. Secondary outcomes included shoulder strength and range of motion (ROM), quality of life (QOL) and physical activity.

**Data availability statement:** All relevant data are within the paper and its Supporting Information files.

**Funding:** This study was funded by an Australian National Health and Medical Research Council Partnership Project Grant (APP1077034). The funding body had no role in the study's design, conduct, analysis, or reporting.

**Competing interests:** The authors have declared that no competing interests exist.

## Results

617 participants were randomly assigned to UL (n = 307) or LL (n = 310) groups. Mean age was 73 years, (SD 6.0) and 64% were female. No clinically important or statistically significant between-group difference was detected in UL function (measured by the DASH) at 12 months (mean difference (MD) = 0.99, 95% CI −0.82 to 2.79, $p$ = 0.283, n = 462). There were no significant between-group differences in shoulder ROM, most measures of strength, physical activity (device-measured and self-report), QOL and UL function at three and six months. Participants performed the exercises twice per week, averaging 104 exercise sessions (SD 69, median 117, range 0–371) over the 12-month intervention period.

## Conclusion

People aged 65 + can successfully learn and adhere to a home-based exercise program for the UL with instruction provided in a group setting, however this program did not improve UL strength, mobility and function. Considering the increased rates of shoulder dysfunction in older age, more research is needed to determine the optimal exercise protocols for prevention of shoulder dysfunction.

## Introduction

Upper limb dysfunction, including shoulder pain and stiffness, is a major health issue for community-living people aged 65 + years [1]. Shoulder pain is the third most common musculoskeletal disorder in adults [2], after back and neck pain [3]. The prevalence of shoulder pain increases with age [4–6], with estimates of point prevalence of shoulder pain in adults aged 70 + years at 13–26% [7]. A 2022 systematic review of the global prevalence and incidence of shoulder pain found prevalence rates ranging from 11–55% for a reference period of 12-months or more, with the annual incidence rate varying between 8–62 per 1000 person years [8]. The economic costs associated with shoulder pain include direct and indirect healthcare costs, work productivity, and personal and domestic support costs [9]. These costs were calculated to be between $13885 and $22378 per year for patients on an orthopaedic waiting list at an Australian public hospital in 2013/14 [9].

The most frequent diagnoses in people with shoulder pain are rotator-cuff tendonitis, subacromial pain syndrome, adhesive capsulitis and acromioclavicular joint disorders [10,11]. However specific diagnoses of shoulder pain are not always straightforward with no clearly defined pathology or physical signs, and may also be the result of coexistence of multiple pathologies, and have therefore been termed as non-specific shoulder pain [11–13].

Hill et al (2010) conducted a population-based survey to estimate the prevalence of shoulder pain and examine its associations with physical function, range of motion and quality of life [6]. The findings revealed that shoulder pain disproportionately affected females and older adults. Specifically, females were 40% more likely than

males to report shoulder pain and/or stiffness, and the likelihood of these symptoms increased with advancing age [6]. Lifestyle factors, including tobacco use, obesity and physical inactivity, were significantly associated with a higher prevalence of shoulder pain and/or stiffness. Individuals experiencing shoulder pain and/or stiffness demonstrated a greater incidence of depressive symptoms and reported lower quality of life scores [6]. Age-related decline in shoulder function was also observed, suggesting potential implications for maintaining independence among older populations.

The consequences of upper limb and shoulder dysfunction are myriad. Shoulder pain creates a significant burden on individuals and the community by reducing an individual's capacity to participate in work (paid and volunteer), recreational activities and activities of daily living [3,12]. Reduced shoulder flexibility also increases the risk of older adults losing social independence [14]. Reduced shoulder movement by older adults significantly reduces daily physical activity, demonstrated by a significant reduction in daily steps while wearing a shoulder orthosis [15]. Shoulder immobilisation also impairs balance in older people, with a study showing a significant reduction in balance ability in community-dwelling people aged 65 years and over wearing shoulder immobilisers [16]. Reduced shoulder range of motion has been identified to be associated with poor lower limb function and mobility in older adults, along with poor walking endurance capacity [17]. Older adults with reduced or abnormal range of motion are 2.5 to 4.5 times more likely to exhibit poor mobility, even after adjusting for age, sex, weight, height and chronic conditions [17]. This information suggests that mobility limitations in older adults can be related to both upper limb and lower limb dysfunction [17].

Preventing and reducing shoulder dysfunction in older adults will have positive impacts on their independence, physical activity, mobility, balance and falls and should be a priority. This is even more imperative when considering global population ageing, which according to the World Health Organisation is occurring at a faster pace than in the past – and 1 in 6 people globally will be aged 60 years and over by 2030 [18]. The significant and growing burden of shoulder pain and disability therefore also has implications for health care costs and systems into the future.

Fortunately, shoulder dysfunction can be successfully managed and treated, and exercise is a common, cost-effective method of treating shoulder disorders [3,19] that has moderate evidence from systematic reviews supporting its efficacy [13,20–23]. A systematic review by Pieters et al (2020) stated that the evidence for exercise as the most important strategy for shoulder pain is increasing [13]. Recent meta-analyses concluded that shoulder specific exercise therapy is effective in reducing chronic shoulder pain and improving function [21,23,24]. Shoulder specific exercises were also more effective for both providing and maintaining pain relief than usual care [24]. Silveira et al (2024) acknowledged that exercise therapy should be used as the first line of treatment in managing chronic shoulder pain [24].

While strengthening exercises in multiple planes of motion focusing on strength and stability of the shoulder joint are recommended [12,21,25,26], a 2024 scoping review of exercise programs for managing rotator-cuff related shoulder pain found a high variability in the parameters used to prescribe exercises for shoulder pain [27]. There is still work to be done to determine if there is an optimal dose and type of exercise for the treatment of shoulder pain.

Similarly, there is limited evidence for the use of upper limb exercise in people with asymptomatic shoulders, with most of the research focused on the treatment of shoulder dysfunction rather than prevention [28]. However, shoulder exercises are routinely provided as part of a general exercise program for older adults [24] and often use elastic exercise bands for resistance, as they are effective in improving strength [29]. A scoping review by Kim et al (2021) of elastic band exercises to improve shoulder function, found only a limited number of studies [28]. It was noted that exercise for shoulder function using elastic bands appeared to have promising results on muscle strength outcomes, however further research is needed to determine how these exercises may affect shoulder function in an older population [28]. One of the goals of the present study, therefore, is to address this notable research gap regarding the preventive effects of strengthening in older adults with asymptomatic shoulders.

While individual physiotherapy exercise programs are the traditional mode of exercise delivery for people with musculoskeletal conditions, group-based exercise programs have been found to be as effective as individual physiotherapy exercise programs [30]. A 2021 study found that group-based exercise, individually supervised exercise and home-based

exercise had similar beneficial effects in people with subacromial pain [31], however the home exercise intervention was shown to be the most cost effective [31]. Granviken and Vasseljen also found similar effects of unsupervised home exercise and supervised exercise on pain and disability for people with subacromial pain when working at the same training dose [32]. Recent systematic reviews and meta-analyses found that home-based exercise alone may be equally effective as other conservative treatments for non-specific shoulder pain [11,23,25], and supervised and unsupervised training (or self-training with one-time instruction) were equally effective on pain and shoulder function [23].

A further consideration related to exercise delivery mode for older people is their varying life circumstances, making a choice of options important [33]. Home-based exercise options may be preferable to older people for reasons of accessibility, lower cost and greater convenience, particularly for those who are unable to leave home [34,35]. This accessibility is particularly important for older adults with mobility limitations, caregiving responsibilities, or those living in regional areas. Home-based exercise programs reduce costs associated with attending sessions and transportation to get to programs [36,37].

The study presented here sits within a larger trial that tested the effects of two different exercise programs in people aged 65 years and over. The results of the lower limb exercise program for falls have been published elsewhere [38]. Building on what is already known about shoulder dysfunction, older people and exercise preferences, the primary aim of the current study was to determine the effect of a home-based exercise program on upper limb function among community-dwelling people aged 65 years and over. Secondary outcomes included shoulder strength and range of motion, quality of life and physical activity.

## Materials and methods

### Study design

A pragmatic randomized controlled trial (RCT) was implemented with two parallel intervention groups. Following baseline assessments, participants were randomly allocated to either the upper limb (UL) intervention or lower limb (LL) control group. The allocation sequence was generated via a computer-based randomization schedule within REDCap, prepared by a researcher who was not involved in participant enrolment. To minimize the risk of contamination, individuals residing in the same household were considered a single unit and assigned to the same intervention group. Given the nature of the exercise interventions, blinding of participants and program facilitators to group assignment was not feasible. However, both primary and secondary outcome data were collected by assessors who were blinded to group allocation, and participants were told to not disclose their assigned exercise group to the assessors. The University of Wollongong and Illawarra Shoalhaven Local Health District Human Research Ethics Committee (HE14/279 and HREC/14/WGONG/50) approved the study. All participants gave written informed consent prior to commencement of data collection. Recruitment occurred between September 2015 and May 2017. Follow-up data collection was completed in May 2018. A comprehensive protocol detailing the study's design and methodology has been published [39]. The study reporting is in accordance with the Consolidated Standards of Reporting (CONSORT) [40,41]. The trial was registered with the Australian and New Zealand Clinical Trials Registry (ACTRN12615000865516) on 19/08/2015, prior to commencement.

### Participants

The study sample comprised community-dwelling individuals aged 65 years and older residing in the Illawarra and Shoalhaven areas, New South Wales, Australia. Recruitment strategies included paid newspaper advertisements, media releases, radio interviews, distribution of printed promotional materials and information sessions delivered to local community groups. Eligibility screening was conducted via telephone interviews. Individuals were excluded if they met any of the following criteria: cognitive impairment (defined by a score below 5 on the Memory Impairment Screen) [42]; unable to walk 10 metres even with the aid of a walking device; insufficient English language proficiency to comprehend program content; diagnosis of a progressive neurological condition; recent fracture or joint replacement within the preceding six

months; presence of a medical condition contraindicating physical activity; unable to obtain clearance to exercise from their General Practitioner; or current engagement in a comparable UL or LL exercise program at a frequency of two or more sessions per week.

### Intervention

The intervention group participants undertook an UL exercise program (BEST at Home – upper limb). The UL exercise program was developed by the research team to improve upper limb strength, mobility, and function. The UL program consisted of one set of eight exercises, which included arm raises, internal and external shoulder rotation, elbow flexion and extension, shoulder press, chest press and shoulder row. Participants received a set of dumbbell weights (ranging from 600g – 3 kg) and an elastic resistance band, available in four levels (light, medium, heavy and extra heavy resistance). The initial exercise load was prescribed by the supervising physiotherapist during the first session. A printed manual outlining the home exercise program, including visual illustrations and detailed instructions for each exercise, was also provided. Participants were instructed to complete 10 repetitions of each exercise, three times per week, in their home setting. All exercises in the UL program were performed while seated. The intervention is outlined using the TIDieR checklist in Table 1 [43]. Participants were taught strategies to progressively increase the difficulty of the exercises as they advanced through the program, such as using a heavier weight or thicker resistance band, increasing the number of sets and/or repetitions of each exercise, increasing the range of motion of some exercises, slowing down the eccentric phase, adding a pause at the point of maximum exertion.

Participants assigned to the LL group engaged in a home-based exercise program designed to reduce fall risk by improving balance and lower limb strength. The program was based on the Otago Exercise Programme [44,45] and comprised of 17 balance and strength exercises. Participants were instructed to complete the exercises at home three times per week [38,39]. The intervention was informed by principles of social cognitive theory [46].

Both the UL and LL exercise interventions were facilitated by two experienced physiotherapists through a series of three group-based workshops. These sessions were held in community venues such as local community centres or clubs, with each group consisting of approximately 10 participants. Workshops were conducted at weeks 1, 4 and 12, each lasting one hour. The exercise programs were individualized to match each participant's level of ability. During each session, exercises were reviewed, techniques refined, and modifications or progressions were made by the physiotherapist based on the participant's performance and needs.

### Outcome measures

**Primary outcome.** The primary outcome for the assessment of the UL intervention was upper limb function measured with the Disability of the Arm, Shoulder and Hand (DASH) questionnaire at 12 months [47]. The DASH is a validated tool that includes 30 items, each rated on a 5-point scale to represent: the difficulty experienced in performing various physical activities that require upper extremity function (physical function, 21 items); symptoms of pain, activity-related pain, tingling, weakness, and stiffness (pain symptoms, 5 items); and impact of disability and symptoms on social activities, work, sleep, and psychological well-being (emotional and social function, 4 items) [47,48]. Scores range from zero to 100, with zero representing no disability and 100 the most severe disability [49]. The DASH has demonstrated good test-retest reliability (ICC 2,1 = 0.93) with sensitivity of 82% and specificity of 74% [50] and a high responsiveness to clinical change [47,51]. A change of approximately 12–14 points on the DASH is generally considered the minimal clinically important difference, representing the smallest change perceived as beneficial by people with upper limb disorders [52].

**Secondary outcomes.** There were several secondary outcomes. Shoulder strength was measured by isometric shoulder internal and external rotation force in both left and right arms using a Lafayette manual muscle tester (Model 01165). Shoulder mobility was measured by active shoulder internal and external rotation range of motion in both left and right arms using a Baseline digital inclinometer (Model 12–1057). The assessments of shoulder strength and mobility were

**Table 1. Intervention description using the template for intervention description and replication (TIDieR) checklist.**

| Checklist item | |
|---|---|
| 1. Brief name | Balance Exercise Strength Training (BEST) at Home (upper limb) trial |
| 2. Why | Upper limb dysfunction, including shoulder pain is a major health issue for older people. Shoulder pain is the third most common musculoskeletal disorder in adults and the prevalence increases with age. |
| 3. What materials | Participants in the upper limb group received:<br>- an exercise program designed to improve upper limb function (including exercise instruction, printed manual, weights and exercise band)<br>Participants in the lower limb group received:<br>- an exercise program designed to reduce falls and improve balance and strength in the lower limbs (including exercise instruction, printed manual and weights)<br>- a brochure on fall prevention titled 'Staying active and on your feet' |
| 4. What procedures | Both intervention groups attended three group-based exercise instruction sessions and three measurement sessions. |
| 5. Who provided | Physiotherapists facilitated the exercise instruction sessions. |
| 6. How | The exercise instruction was provided in person in small groups of approximately 10 participants. |
| 7. Where | In the Illawarra and Shoalhaven regions, New South Wales, Australia. |
| 8. When and how much | Exercise instruction sessions were facilitated in weeks 1, 4 and 12. Each session was one hour duration. Participants were asked to perform the exercises three times per week for 12 months. |
| 9. Tailoring | The exercises were individualized by the physiotherapist for each participant, based on their level of ability and need. |
| 10. Modifications | No modifications were made. |
| 11. How well (planned) | Adherence to the exercise program was determined by self-reported exercise sessions, which were marked on postal calendars (and returned monthly) |
| 12. How well (actual) | Participants were instructed to complete the exercises 3 times per week.<br>Participants in the upper limb group completed an average of 104 sessions over the 12-month period (twice per week).<br>Participants in the lower limb group completed an average of 94 sessions over the 12-month period (less than twice per week). |

conducted at baseline, three months and six months by physiotherapists who were blinded to group allocation. Quality of life was assessed with the self-report Short Form Survey 12, version 2 (SF12v2) [53]. Physical activity (including daily step count, counts per minute and minutes of moderate to vigorous physical activity) was measured with an Actigraph accelerometer (model wGT3x-BT) worn at the waist [54,55]. Accelerometer data were collected over a one-week period to account for day-to-day variation in physical activity levels. Acceptable wear time was defined as a minimum of four days of 10 hours or more per day. Physical activity was also measured using self-report data from the Incidental and Planned Exercise Questionnaire [56]. Paper-based questionnaires were self-completed during sessions at baseline, 12 weeks and 6 months; and via postal questionnaires at 12 months post-randomisation.

Participants also answered questions at baseline about sociodemographic details, prescription medication and comorbidities. To measure program adherence, participants were asked to record the days that they completed the exercises on the calendars that were returned each month.

## Data analysis

A pre-specified statistical analysis plan was followed. Linear regression models were used to assess the effect of group allocation on the continuously scored measures (upper limb function via the DASH questionnaire, shoulder internal and external rotation force and range of movement, physical activity, quality of life), after adjusting for baseline scores [57]. For variables that were not normally distributed (DASH score, SF-12, moderate-vigorous physical activity (MVPA), planned physical activity, total walking) change scores from baseline to follow-up were analysed. Separate linear regression analyses were performed for each time point for continuous measures. Interaction terms in the model were used to explore potential differential intervention effects according to age, sex (male versus female), previous falls and baseline DASH score. All statistical tests were two-tailed with statistical significance set at $p < 0.05$. No formal adjustment for multiple comparisons was undertaken. The data analyses were undertaken blinded to group allocation and used an intention-to-treat approach. Sample size calculations suggested that 576 participants would be sufficient to detect a 10% between-group difference in the DASH total score and the secondary physical outcomes, with a 15% loss to follow up [39]. The sample size calculation used the nbpower user written command in Stata (Stata Statistical Software: Release 15. College Station, TX: StataCorp LLC). A sensitivity analysis was conducted for the primary outcome to account for clustering of household participants.

## Results

### Participants

The participant flow throughout the trial is displayed in Fig 1. Of the 953 individuals assessed for eligibility, 308 declined participation and 28 did not satisfy the inclusion criteria. A total of 617 participants (mean age 73 years, SD 6, 64% female) were randomly assigned to either the UL group (n = 310) or LL group (n = 307). Baseline characteristics are detailed in Table 2, with both groups being well-matched at baseline (Table 2).

### Effect of intervention

**Primary outcome.** No clinically important or statistically significant between-group difference was detected in UL function (as measured by the DASH questionnaire) at 12 months (mean difference = 0.99, 95% CI −0.82 to 2.79, $p = 0.283$, n = 462). A sensitivity analysis conducted to adjust for clustering of households found similar results, with no difference in UL function in the UL group compared to the LL group (mean difference = 0.99, 95% CI −0.84 to 2.82, $p = 0.289$, n = 462).

**Secondary outcomes.** Table 3 shows the baseline and follow-up scores for the secondary outcomes. There was a significant between-group difference in shoulder strength, favouring the LL group, for both internal rotation (mean difference = −0.58, 95% CI −1.08 to −0.09, $p = 0.021$) and external rotation (mean difference = −0.51, 95% CI −0.95 to −0.06, $p = 0.026$) on the left side at three months but not at six months, however these differences were unlikely to be clinically important. There were no between-group differences in shoulder range of motion, physical activity (device-measured and self-report), quality of life and UL function at 3 or 6 months.

### Sub-group analyses for the primary outcome

In planned sub-group analyses, no statistically significant differences in the intervention's effect on the primary outcome of UL function by age ($p$ for interaction = 0.933), sex (male versus female, $p = 0.262$), or upper limb dysfunction at entry to the trial, determined by a DASH score >15 at baseline ($p = 0.932$) [58] were observed. Intervention effectiveness varied

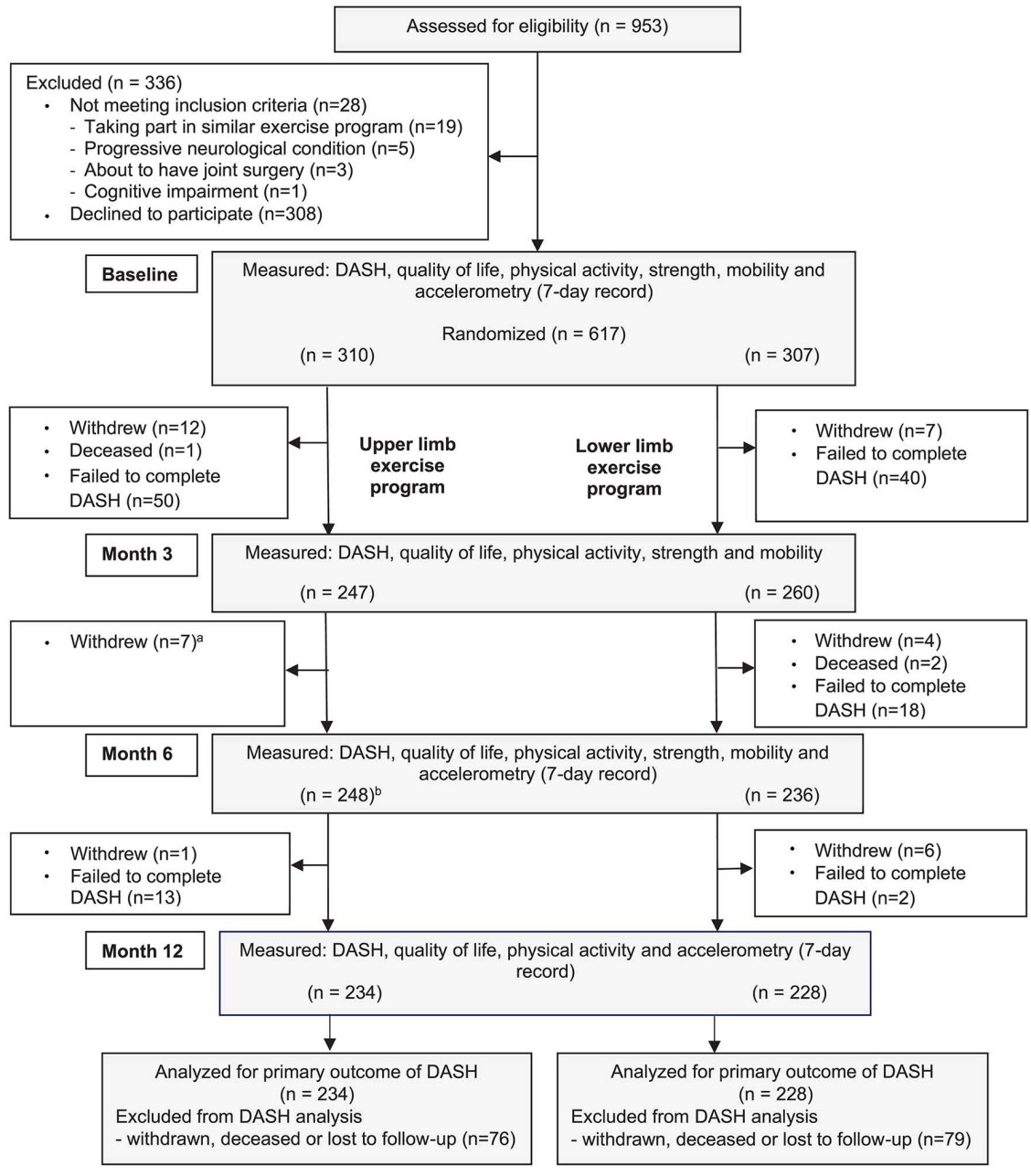

**Fig 1. Design and flow of participants through the trial.** [a] These participants withdrew from the study but still completed the questionnaire and their data were included in the analysis. [b] One participant who had failed to complete three-month questionnaire completed the six-month questionnaire. DASH = Disability of the Arm, Shoulder and Hand questionnaire.

significantly in people who had and had not fallen in the 12 months prior to baseline ($p = 0.027$). For the subgroup of participants who had fallen in the past 12 months, the UL group displayed better UL function (lower DASH scores, mean 15.6, SD 14.8), than the LL group (DASH mean 19.1, SD 14.9) ($p = 0.201$) at 12 months. However, for participants who had not fallen in the 12 months prior to baseline, the LL group displayed better UL function (DASH mean 10.2, SD 12) compared

**Table 2. Characteristics of participants at baseline (n = 617).**

| Characteristics | Upper limb | n | Lower limb | n | Total | n |
|---|---|---|---|---|---|---|
| Age (years), mean (SD) | 73.2 (5.8) | 310 | 72.9 (6.2) | 307 | 73.1 (6.0) | 617 |
| Female sex: n (%) | 197 (64) | 310 | 196 (64) | 307 | 393 (64) | 617 |
| Lives alone: n (%) | 97 (31) | 310 | 89 (29) | 307 | 186 (30) | 617 |
| Medical conditions (0–17)[a], mean (SD) | 2.8 (2.0) | 310 | 2.8 (1.9) | 307 | 2.8 (1.9) | 617 |
| Arthritis: n (%) | 181 (59) | 309 | 173 (58) | 297 | 354 (58) | 606 |
| Osteoporosis: n (%) | 57 (19) | 307 | 66 (22) | 304 | 123 (20) | 611 |
| Diabetes: n (%) | 32 (10) | 307 | 26 (9) | 306 | 58 (10) | 613 |
| Depression: n (%) | 48 (16) | 303 | 51 (17) | 303 | 99 (16) | 606 |
| Total medications (n), mean (SD) | 3.3 (2.7) | 305 | 3.0 (2.6) | 304 | 3.1 (2.7) | 609 |
| DASH outcome measure, mean (SD) | 13.2 (12.1) | 310 | 13.2 (11.9) | 307 | 13.2 (12.0) | 617 |
| DASH score > 15[b]: n (%) | 104 (34) | 310 | 108 (35) | 307 | 212 (34) | 617 |
| SF12v2: physical composite score, mean (SD) | 47.8 (7.4) | 298 | 48.0 (7.7) | 304 | 47.9 (7.6) | 602 |
| SF12v2: mental composite score, mean (SD) | 53.6 (5.4) | 298 | 53.4 (5.7) | 304 | 53.5 (5.6) | 602 |
| Average daily step count, steps, mean (SD) | 5539.6 (2394.5) | 288 | 5725.9 (2424.6) | 297 | 5634.2 (2409.5) | 585 |
| Self-report physical activity, hours/week[c]: mean (SD) | 34.3 (19.2) | 309 | 33.4 (19.1) | 306 | 33.9 (19.1) | 615 |

[a]Possible medical conditions included: arthritis, osteoporosis, asthma, chronic obstructive pulmonary disease, angina, heart disease, heart attack, neurological disease, stroke/transient ischemic attack, peripheral vascular disease, diabetes mellitus, upper gastrointestinal disease, depression, anxiety/panic disorder, visual impairment, hearing impairment, degenerative disc disease)

[b]DASH score >15 indicates some upper limb dysfunction

[c]Measured using the Incidental and Planned Exercise Questionnaire

DASH: Disabilities of the Arm, Shoulder, and Hand questionnaire

to the UL group (mean 13.2, SD 15) (p = 0.035). However, these subgroup findings should be interpreted with caution as the trial was not powered to detect subgroup interactions, and these analyses were exploratory in nature.

## Adherence with the program

Attendance at the group-based exercise instruction sessions was documented. In the UL group, 298 (96%) participants were present at the week 1 session, 260 (84%) and 241 (78%) were present at weeks 4 and 12 respectively. In the LL group, 294 (96%) participants were present at the week 1 session, 260 (85%) and 242 (79%) were present at weeks 4 and 12 respectively. A total of 280 participants (90%) in the UL group and 279 participants (91%) in the LL group attended at least two of the three exercise instruction sessions.

Participants in the UL group reported performing the exercises twice per week, averaging 104 exercise sessions (SD 69, median 117, range 0–371) over the 12-month intervention period. In contrast, participants in the LL group reported exercising less than twice per week, with a mean of 94 exercise sessions (SD 63, median 97, range 0–287). On average, participants submitted calendar data for 10 months, and 424 participants (69%) returned completed calendars for all 12 months.

One participant in the UL group experienced a minor adverse event when an exercise band broke, requiring them to discontinue the program and seek medical attention. Additionally, 23 participants (12 UL intervention group, 11 LL control group) reported minor musculoskeletal discomfort, all of which resolved within a short duration.

## Acceptability of the intervention

Participants perceptions of the intervention were assessed through surveys administered at 3, 6 and 12 months. Response rates differed across assessment periods. Confidence in completing the home-based exercises was reported

**Table 3. Intervention effects on primary and secondary outcomes.**

| Outcome measure | Upper limb Mean (SD), n | Lower limb Mean (SD), n | Mean difference (95% CI) | P value |
|---|---|---|---|---|
| **Primary outcome** | | | | |
| **DASH score [a][c]** | | | | |
| Baseline | 13.2 (12.1) n=307 | 13.2 (11.9) n=307 | | |
| 3 months | 12.3 (12.1) n=247 | 12.5 (12.5) n=260 | 0.39 (−0.93-1.71) | 0.559 |
| 6 months | 13.1 (13.4) n=248 | 12.7 (12.8) n=236 | 0.11 (−1.38-1.60) | 0.882 |
| 12 months | 13.8 (14.9) n=234 | 12.4 (13.3) n=228 | 0.99 (−0.82-2.79) | 0.283 |
| **Secondary outcomes** | | | | |
| **Shoulder strength, internal rotation force, right, kg, mean (SD) [b]** | | | | |
| Baseline | 10.2 (4.5) n=309 | 10.6 (5.0) n=307 | | |
| 3 months | 10.8 (4.7) n=238 | 11.4 (4.9) n=244 | −0.28 (−0.76-0.19) | 0.244 |
| 6 months | 11.5 (5.0) n=212 | 11.3 (5.0) n=209 | 0.55 (−0.04-1.13) | 0.069 |
| **Shoulder strength, internal rotation force, left, kg, mean (SD) [b]** | | | | |
| Baseline | 10.0 (4.6) n=309 | 10.3 (4.6) n=307 | | |
| 3 months | 10.3 (4.6) n=239 | 11.1 (4.9) n=243 | −0.58 (−1.08--0.09) | 0.021* |
| 6 months | 11.0 (4.9) n=213 | 11.1 (4.9) n=207 | −0.02 (−0.59-0.55) | 0.950 |
| **Shoulder strength, external rotation force, right, kg, mean (SD) [b]** | | | | |
| Baseline | 9.0 (4.4) n=309 | 9.4 (5.2) n=307 | | |
| 3 months | 9.8 (4.4) n=238 | 10.3 (4.7) n=244 | −0.37 (−0.82-0.07) | 0.102 |
| 6 months | 10.7 (5.0) n=212 | 10.7 (5.3) n=209 | 0.28 (−0.36-0.91) | 0.395 |
| **Shoulder strength, external rotation force, left, kg, mean (SD) [b]** | | | | |
| Baseline | 9.4 (4.3) n=309 | 9.8 (4.8) n=307 | | |
| 3 months | 9.7 (4.2) n=239 | 10.7 (4.6) n=243 | −0.51 (−0.95--0.06) | 0.026* |
| 6 months | 10.4 (4.7) n=213 | 10.8 (5.0) n=207 | 0.12 (−0.46-0.69) | 0.692 |
| **Shoulder internal rotation range of motion, right, degrees, mean (SD) [b]** | | | | |
| Baseline | 65.1 (14.6) n=309 | 65.8 (14.1) n=307 | | |
| 3 months | 64.7 (13.6) n=238 | 66.8 (14.6) n=244 | −1.67 (−3.75-0.41) | 0.115 |
| 6 months | 65.9 (14.7) n=213 | 66.7 (14.6) n=209 | −0.50 (−2.95-1.94) | 0.686 |
| **Shoulder internal rotation range of motion, left, degrees, mean (SD) [b]** | | | | |
| Baseline | 70.1 (13.6) n=309 | 70.8 (13.0) n=307 | | |
| 3 months | 66.9 (13.6) n=239 | 69.2 (14.2) n=243 | −1.90 (−3.98-0.18) | 0.074 |
| 6 months | 67.0 (14.8) n=213 | 68.1 (14.7) n=209 | −0.51 (−3.05-2.02) | 0.690 |
| **Shoulder external rotation range of motion, right, degrees, mean (SD) [b]** | | | | |
| Baseline | 74.2 (16.8) n=309 | 74.0 (15.7) n=307 | | |
| 3 months | 74.5 (14.8) n=238 | 73.5 (16.0) n=244 | 0.44 (−1.59-2.46) | 0.672 |
| 6 months | 74.4 (16.4) n=213 | 72.7 (17.2) n=209 | 1.32 (−1.04-3.69) | 0.272 |
| **Shoulder external rotation range of motion, left, degrees, mean (SD) [b]** | | | | |
| Baseline | 70.6 (15.7) n=309 | 71.9 (16.3) n=307 | | |
| 3 months | 71.6 (15.0) n=239 | 73.5 (16.7) n=243 | −1.10 (−3.14-0.93) | 0.287 |
| 6 months | 72.4 (15.7) n=213 | 72.7 (18.4) n=209 | 0.85 (−1.37-3.07) | 0.451 |
| **SF12 physical component summary score [b][c]** | | | | |
| Baseline | 47.8 (7.4) n=298 | 48.0 (7.7) n=304 | | |
| 3 months | 48.1 (7.0) n=243 | 48.5 (7.6) n=257 | −0.28 (−1.2-0.6) | 0.534 |
| 6 months | 47.7 (7.8) n=244 | 48.0 (8.2) n=238 | 0.26 (−0.8-1.3) | 0.633 |
| 12 months | 47.2 (8.7) n=228 | 48.5 (7.6) n=227 | −1.1 (−2.3-0.008) | 0.052 |

*(Continued)*

**Table 3.** (Continued)

| Outcome measure | Upper limb Mean (SD), *n* | Lower limb Mean (SD), *n* | Mean difference (95% CI) | *P* value |
|---|---|---|---|---|
| **SF12 mental component summary score** [b c] | | | | |
| Baseline | 53.6 (5.4) n = 298 | 53.4 (5.7) n = 304 | | |
| 3 months | 54.4 (5.3) n = 243 | 54.1 (5.6) n = 257 | −0.1 (−1.1-0.8) | 0.808 |
| 6 months | 54.1 (5.1) n = 244 | 54.2 (5.3) n = 238 | −0.7 (−1.6-0.3) | 0.185 |
| 12 months | 54.2 (4.8) n = 228 | 54.4 (5.0) n = 227 | −0.1 (−1.1-0.8) | 0.824 |
| **Physical activity, accelerometer (counts per minute)** [b] | | | | |
| Baseline | 230 (103) n = 288 | 239 (111) n = 297 | | |
| 6 months | 238 (113) n = 236 | 246 (111) n = 230 | 0.4 (−14.1-14.8) | 0.960 |
| 12 months | 246 (119) n = 193 | 249 (111) n = 205 | 3.2 (−10.7-17.0) | 0.653 |
| **Daily steps *(n)*** [b] | | | | |
| Baseline | 5540 (2394) n = 288 | 5726 (2425) n = 297 | | |
| 6 months | 5689 (2337) n = 236 | 5957 (2653) n = 231 | −105 (−429−219) | 0.525 |
| 12 months | 5916 (2638) n = 193 | 5958 (2532) n = 205 | 90 (−223-402) | 0.572 |
| **Moderate-vigorous physical activity, minutes/day, mean (SD)** [b c] | | | | |
| Baseline | 17.2 (16.8) n = 288 | 18.9 (17.4) n = 297 | | |
| 6 months | 17.6 (17.1) n = 236 | 20.0 (18.1) n = 231 | −0.4 (−3.1-2.3) | 0.775 |
| 12 months | 19.0 (18.6) n = 193 | 19.8 (18.0) n = 205 | 0.3 (−2.1-2.8) | 0.785 |
| **IPEQ, total physical activity, hours per week, mean (SD)** [b] | | | | |
| Baseline | 34.3 (19.2) n = 309 | 33.4 (19.1) n = 306 | | |
| 3 months | 33.1 (18.1) n = 247 | 33.4 (19.4) n = 260 | −1.1 (−3.6-1.5) | 0.412 |
| 6 months | 33.0 (17.6) n = 250 | 31.7 (17.3) n = 244 | 0.9 (−1.7-3.5) | 0.494 |
| 12 months | 32.5 (17.6) n = 235 | 31.3 (16.7) n = 229 | 0.9 (−1.7-3.6) | 0.491 |
| **Planned physical activity (excluding walking), hours per week, mean (SD)** [b c] | | | | |
| Baseline | 2.2 (3.4) n = 310 | 2.3 (3.8) n = 307 | | |
| 3 months | 2.7 (3.1) n = 251 | 2.8 (3.8) n = 262 | −0.2 (−0.8-0.5) | 0.575 |
| 6 months | 2.4 (2.9) n = 250 | 2.7 (3.6) n = 244 | −0.04 (−0.7-0.6) | 0.895 |
| 12 months | 2.1 (3.5) n = 237 | 2.1 (3.1) n = 234 | 0.2 (−0.4-0.9) | 0.514 |
| **Total walking, hours per week, mean (SD)** [b c] | | | | |
| Baseline | 4.6 (4.8) n = 310 | 4.7 (5.6) n = 307 | | |
| 3 months | 5.0 (4.7) n = 249 | 5.3 (6.7) n = 261 | −0.2 (−1.2-0.8) | 0.702 |
| 6 months | 4.8 (5.1) n = 250 | 4.5 (4.2) n = 242 | 0.4 (−0.5-1.3) | 0.350 |
| 12 months | 5.0 (6.1) n = 237 | 5.1 (5.6) n = 234 | −0.03 (−1.1-1.0) | 0.951 |
| **Incidental physical activity (including walking), hours per week, mean (SD)** [b] | | | | |
| Baseline | 29.3 (17.7) n = 309 | 28.4 (18.2) n = 306 | | |
| 3 months | 27.4 (16.6) n = 246 | 27.7 (17.4) n = 261 | −0.8 (−3.1-1.6) | 0.519 |
| 6 months | 28.2 (16.9) n = 246 | 26.2 (16.6) n = 243 | 1.5 (−1.0-4.0) | 0.245 |
| 12 months | 27.3 (16.2) n = 235 | 25.9 (16.0) n = 229 | 0.87 (−1.6-3.3) | 0.487 |

[a]Lower scores reflect better performance

[b]Higher scores reflect better performance

[c]Skewed distribution

[*]Significant outcome

DASH: Disabilities of the Arm, Shoulder, and Hand questionnaire

IPEQ: Incidental and Planned Exercise Questionnaire

by 98% of participants in the upper limb group (239/245) at 3 months. At 6 months, 95% (236/248) of UL group participants expressed their intent to continue the exercise program. The 12-month assessment yielded a mean perceived benefit rating of 7.5 out of 10 (SD 2.1) for the UL group. At this timepoint, 81% of participants (190/235) planned to continue the exercises, and 92% (217/236) stated they would recommend the program to peers aged 65 years or older.

Intervention group participants reported their preferred aspects of the program at 12 months. The highest rated features were the ability to exercise at home (95%, n = 222/234), the simple nature of the exercises (94%, n = 221/234) and the flexibility to perform the exercises at any time (93%, n = 218/234). Despite these positive features, nearly half (46%, 109/237) UL group participants experienced challenges with regular exercise completion. The most frequently reported barriers included injury (17%, n = 40/237), family obligations (15%, n = 36/237), lack of motivation (15%, n = 36/237), travel (14%, n = 32/237), time constraints (11%, n = 26/237), and health issues (11%, n = 25/237).

## Discussion

### Overall findings

Our study found that this home-based exercise program was not effective at improving upper limb strength, mobility and function in community-dwelling people aged 65 years and over. There was no difference in the primary outcome of the DASH score measured at 12 months post-randomisation between the UL and LL groups. There were no significant between-group differences in shoulder range of motion, most measures of shoulder rotation strength, physical activity, quality of life and UL function at three and six months.

Given that exercises used in the UL intervention group are commonly prescribed and used in other studies to improve shoulder function and reduce pain in adults [19,21,59] it was somewhat surprising to find these exercises were not effective in improving shoulder function, strength or ROM, in community-dwelling people aged 65 years and over in this study, when compared to a control group. There are several possibilities for these findings.

Firstly, there was a low level of supervision and supervised exercise progression in this study, with only three sessions of exercise instruction in the first three months of the 12-month intervention period. While participants were given suggestions on how to progress the exercises (e.g., heavier weights, more repetitions and increasing exercise complexity), supervised exercise progression beyond this time was not offered. This may have provided insufficient challenge to strength throughout the 12-month follow-up. Intervention group participants completed the exercises on average twice weekly over the 12-month period, potentially representing an inadequate exercise dose to improve upper limb function and strength. In addition to unsupervised exercise providing insufficient challenge or motivation, it is also possible that incorrect performance of the exercises did not have the desired effect. When participating in group classes, posture and position can be regularly checked and feedback provided, which may also influence outcomes [60].

Secondly, unlike previous studies [31,32] participants in this study were not recruited specifically because of any pre-existing upper limb dysfunction or pain. Motivation may have been lower in those without upper limb dysfunction, as they may not have prioritised the need for such an exercise program. Indeed, some participants expressed a preference for the LL exercise program which had a focus on exercises to prevent falls, and this may have altered their motivation to continue with the UL exercises. This was not, however, reflected in their adherence to the program when compared with the LL intervention, which was roughly the same.

A significant improvement in the LL group compared with the UL group was found for shoulder internal and external rotation strength on the left side, at three months. This unexpected finding may have been due to the LL group steadying themselves with the left arm while performing the LL exercises. This improvement identified only at the 3-month time point, could indicate that participants in the LL group were holding on to stabilise themselves during the balance exercises. Participants in the LL group were encouraged to reduce their support (holding on) during the balance exercises as a progression and therefore may not have been holding for support as their balance and confidence improved throughout the later part of the program. Although no statistically significant between-group differences were observed, these findings

should not be interpreted as evidence that the intervention has no effect, rather, the results indicate that this study did not detect a statistically significant difference between groups.

The intervention's strong acceptability indicates that adults aged 65 and over can successfully learn to perform a home-based exercise program through group-based instruction. The majority of participants found the UL exercises easy to follow and appreciated the flexibility of completing them at home according to their own schedule. Even though our program did not have the social support that group-based programs offer, participants had more control over when the exercises were completed, which has been identified as a facilitator to physical activity and program participation [61,62].

Despite this, almost half of the participants identified obstacles to consistent exercise completion. These included injury, family obligations, lack of motivation, travel, being too busy and health issues. Such barriers are frequently documented in existing literature [61,62] and highlight challenges related to the feasibility of sustaining regular exercise in this population. Future interventions aiming to increase exercise completion may benefit from incorporating targeted behavior change strategies, such as supportive phone calls, text messages, online videos, peer meetings or mobile device applications.

### Strengths

This study possessed numerous strengths. It employed a pragmatic RCT design, with inclusive eligibility criteria. It adhered to Consolidated Standards of Reporting Trials (CONSORT) guidelines [63] and had a prospectively registered and published protocol [39]. Bias mitigation strategies included concealed randomization blinded outcome assessment. Although the data for the primary outcome of upper limb function was participant-reported using the DASH questionnaire, the staff member collecting this information remained blinded to group allocation, and all data handling was conducted in a blinded fashion. The data for the secondary outcomes were measured by trained health professionals and validated tools were used to assess physical activity and quality of life. The program demonstrated strong acceptability, with 81% of respondents expressing intent to continue the exercises and 93% would recommend it to peers aged 65+.

### Limitations

The study had several limitations. Suboptimal program adherence may have prevented participants from achieving the recommended strength training dose, with participants on average completing the exercises twice per week. There may not have been enough instruction sessions later in the program to allow the exercises to be progressed sufficiently. Declining attendance at later instruction sessions may have further compromised exercise intensity, as these workshops specifically taught participants how to advance the exercises and increase difficulty. The intervention was intentionally designed as a pragmatic trial with minimal participant contact to reflect a scalable community-based program. While participants were provided with guidance on progression, the relatively low level of supervision may have limited the extent to which some individuals progressed exercise intensity sufficiently to elicit measurable physiological adaptations. It is therefore possible that a higher training dose, more structured progression, or greater supervision may be required to optimise strength and functional improvements in older adults.

The cohort consisted of reasonably fit and healthy older adults who voluntarily responded to advertising. Participants were not recruited with a focus on pre-existing upper limb pain or dysfunction, and therefore some of the exercises may not have been challenging enough and it is possible that there wasn't much improvement to gain from their baseline level of function and DASH score. This low level of UL dysfunction at entry to the study may have been a limitation to detecting an improvement in UL function and change may have been detected if a more sensitive, high level UL strength test (e.g., 1RM) was the primary outcome measure.

### Conclusion

These findings indicate that additional research is needed to determine whether this type of home exercise program could be beneficial to older adults in general, or if it would provide more benefit to people with pre-existing UL dysfunction. A

recommendation for future research is to investigate combining an UL program with a LL program for a more comprehensive home-based exercise program to improve overall strength and functional independence in people 65+ while meeting the need for people who cannot access exercise programs or prefer to exercise in their own home environment. Considering the increased rates of shoulder dysfunction in older age groups, and the emerging evidence of the impact of shoulder dysfunction on mobility limitations among older adults [17], more research is needed to determine the optimal exercise protocols for prevention of shoulder dysfunction in healthy older adults.

## Supporting information

**S1 File. CONSORT_2025_editable_checklist_Bates_BEST_UL_2025-10-22.**
(DOCX)

**S1 Data. Final 2021-02-22_for PLOS One - May 2026.**
(XLSX)

## Acknowledgments

The authors are grateful to the study participants; Vanessa Jackson, who provided valuable administrative support; and Physiotherapy, Exercise Physiology and Health Promotion staff from Illawarra Shoalhaven Local Health District who assisted in the collection of data and instruction of exercises.

## Author contributions

**Conceptualization:** Amanda Bates, Cathie Sherrington, Susan Furber, Paul van den Dolder, Karen Ginn, Adrian Bauman, Michelle Kershaw, Anne Tiedemann.

**Data curation:** Amanda Bates.

**Formal analysis:** Amanda Bates, Cathie Sherrington, Anne Tiedemann.

**Funding acquisition:** Amanda Bates, Cathie Sherrington, Susan Furber, Anne Tiedemann.

**Investigation:** Amanda Bates, Susan Furber, Michelle Kershaw, Cathy Chittenden.

**Methodology:** Amanda Bates, Cathie Sherrington, Susan Furber, Paul van den Dolder, Karen Ginn, Adrian Bauman, Michelle Kershaw, Anne Tiedemann.

**Project administration:** Amanda Bates, Susan Furber, Lisa Franco, Cathy Chittenden, Anne Tiedemann.

**Resources:** Amanda Bates, Cathie Sherrington, Susan Furber, Karen Ginn, Michelle Kershaw, Lisa Franco, Cathy Chittenden, Anne Tiedemann.

**Supervision:** Cathie Sherrington, Susan Furber, Heidi Gilchrist, Anne Tiedemann.

**Writing – original draft:** Amanda Bates, Susan Furber, Anne Tiedemann.

**Writing – review & editing:** Amanda Bates, Cathie Sherrington, Susan Furber, Heidi Gilchrist, Paul van den Dolder, Karen Ginn, Adrian Bauman, Michelle Kershaw, Lisa Franco, Cathy Chittenden, Anne Tiedemann.

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
