## [Decision Letter · Decision Letter 0]

4 Feb 2026

PONE-D-25-56497Effectiveness of a home-based exercise program for improving upper limb function in community-dwelling older people: a pragmatic randomized controlled trialPLOS One

Dear Dr. Bates,

Thank you for submitting your manuscript to PLOS ONE. After careful consideration, we feel that it has merit but does not fully meet PLOS ONE’s publication criteria as it currently stands. Therefore, we invite you to submit a revised version of the manuscript that addresses the points raised during the review process.

We look forward to receiving your revised manuscript.

Kind regards,

Gurkan Gunaydin

Academic Editor

PLOS One

**Journal Requirements:**

2. We note that your Data Availability Statement is currently as follows:

“All relevant data are within the manuscript and its Supporting Information files.”

3. Please upload a new copy of Figure 1  as the detail is not clear. Please follow the link for more information:  https://journals.plos.org/plosone/s/figures

**Additional Editor Comments:**

Dear authors, please provide a point-by-point response to the reviewers' comments

Reviewers' comments:

Reviewer's Responses to Questions

**Comments to the Author**

1. Is the manuscript technically sound, and do the data support the conclusions?

Reviewer #1: Yes

Reviewer #2: Yes

2. Has the statistical analysis been performed appropriately and rigorously? 

Reviewer #1: Yes

Reviewer #2: Yes

3. Have the authors made all data underlying the findings in their manuscript fully available?

Reviewer #1: Yes

Reviewer #2: Yes

4. Is the manuscript presented in an intelligible fashion and written in standard English?

Reviewer #1: Yes

Reviewer #2: Yes

5. Review Comments to the Author

Reviewer #1: GENERAL COMMENTS

Thank you for the opportunity to review manuscript # PONE-D-25-56497) titled “Effectiveness of a home-based exercise program for improving upper limb function in

community-dwelling older people: a pragmatic randomized controlled trial”. Overall, I think this is an important and well-done study and I enjoyed reading it. I am especially encouraged that the authors submitted this manuscript despite what were essentially null results. Why? Two reasons – First, the quality of any manuscript should be driven by the quality of the methods and not the results. Second, knowing what might not work, as well as what does work, are both important.

SPECIFIC COMMENTS

*Page 3, lines 53 and 54 (Abstract) – Since you say here in your Conclusions that people “can successfully learn and adhere to a home-based exercise...” I would suggest that you briefly report your results for adherence somewhere in the Results section of your abstract.

*Page 4 (Introduction) – I think this is a nice introduction. I only have one suggestion. Somewhere in your Introduction, briefly describe, with appropriate citations, the economic costs associated with upper limb dysfunction. This would provide additional justification for your study.

*Pages 8 through 14 (Materials and Methods) – In either a separate section at the end of your Methods or throughout your Methods section, please describe any post hoc changes to your a priori protocol, including the reasons for those changes.

*Page 9, lines 195 and 196 (Materials and Methods, Study Design) – Since you followed the CONSORT guidelines, and in addition to reference #39 (CONSORT 2010 Statement), it would be better to also include the CONSORT guidelines extension statement for pragmatic trials. Here’s the reference: Zwarenstein M, Treweek S, Gagnier JJ, et al. Improving the reporting of pragmatic trials: an extension of the CONSORT statement. BMJ. 2008;337:a2390. Doi: 10.1136/bmj.a2390. This may also be helpful to you in the reporting of your entire study.

*Pages 10 through 12, lines 217 through 252 (Materials and Methods, Intervention) – Please specify the number of sets of each exercise that participants performed at each session. I’m assuming one but am not sure. Assuming this is accurate, maybe on lines 219 and 220 something like …”consisted of one set of eight exercises,… versus “…”consisted of a set of eight exercises,…” Also, on lines 230 and 231 could you please describe the “strategies to progressively increase the difficulty of the exercises as they advanced through the program.” I think some readers, including myself, would be especially interested in this. Finally, on lines 242 and 243, I’m encouraged to see you mention the behavioral intervention that you used.

*Pages 12 and 13 (Materials and Methods, Outcome measures, Primary outcome, Secondary outcomes) – In my opinion, a nice description of your primary and secondary outcomes.

*Page 14 (Materials and Methods, Data analysis) – Overall, a nice data analysis section. However, while you mention clinical relevance back in your abstract as well as in your Results, I do not see any information regarding what you considered to be a minimal clinically important difference (MCID) for your primary and secondary outcomes. I do however see data regarding a statistically significant difference for your primary and secondary physical outcomes, i.e., “…10% between-group difference in the DASH total score and the secondary physical outcomes, with a 15% loss to follow up (38).” Please elaborate. Also, please describe exactly what linear regression models you used with appropriate citations where necessary. Furthermore, please tell the reader if any adjustment was made for the multiple statistical tests you conducted and how this was done. If no adjustments were made, then this should also be stated. Finally, on line 303, please tell the reader if your tests for statistical significance were one-tailed or two-tailed.

*Page 15 (Table 2) – It would be nice to see 95% confidence intervals for this data.

*Page 17 (Results, Effect of intervention, Secondary outcomes) – Please tell the reader if the differences you observed were considered to be clinically important.

*Page 18 (Table 3) – In addition to absolute difference, please also report relative, i.e., percent, changes.

*Page 20 (Results, Adherence with the program) – Overall, a very nice description of adherence. I just have one suggestion. In the first paragraph, suggest that you report 95% confidence intervals for your percent adherence data.

*Page 21 (Acceptability of the intervention) – Overall, a very nice description here. Along those lines, it does not appear that kinesiophobia is a major issue for these folks.

*Pages 21 through 25 (Discussion) – Overall, a nice Discussion. However, for ease of reading, I would suggest that you partition your Discussion into the following subsections and address: (1) Overall findings, (2) Implications for research, (3) Implications for practice, (4) Implications for policy, if any, and (5) Strengths and potential limitations. Also, I think it is important to point out somewhere that “absence of evidence” is not “evidence of absence”.

END OF REVIEW

Reviewer #2: Thank you for the opportunity to review this manuscript.

A few points to consider

Introduction

- When referring to articles consider adding the date ex: Kim et al., (XXXX)

- I think the introduction can be made more concise. Consider condensing the concepts of shoulder dysfunction as well as condense your rationale for exercise interventions in the shoulder. Refer to systematic reviews and meta-analyses were appropriate to help summarize the literature more concisely.

- Maybe highlight the literature around the DASH to help justify it’s use as your primary outcome

Methods:

- I wonder if the Otago program was challenging enough for this demographic for the LL group, maybe biasing the upper extremity group

- Include a sample size calculation

- Lines ~217–233, 388–403

• The intervention relies on limited supervision (three sessions over 12 months).

• Average adherence (~2 sessions/week) falls below the prescribed dose.

• Exercises were performed seated, with modest resistance

• Clarify whether the intervention was designed as maintenance, prevention, or strength-building, and explicitly discuss whether the dose and progression were sufficient to elicit physiological adaptation in older adults.

- Strength and ROM were assessed only to 6 months, whereas DASH extended to 12 months.

Results:

- Lines ~349–364

• Results focus on statistical significance without adequate discussion of clinical relevance, particularly given low baseline impairment.

- Lines ~375–386

• The interaction with prior falls is interesting but underpowered and exploratory.

• Interpretation risks overstating importance.

Discussion

- Lines ~432–458 (Initial interpretation of findings)

• The discussion attributes null findings largely to adherence and supervision but does not sufficiently consider measurement sensitivity, baseline function, or comparator effects.

- Lines ~527–539

• Statements risk being interpreted as “home-based UL exercise is ineffective,” which is broader than supported by the data.

Minor Suggestions:

- Terms such as upper limb dysfunction, shoulder dysfunction, and shoulder pain are used interchangeably.

- In the discussion more explicitly distinguish acceptability, feasibility, and effectiveness to avoid conceptual overlap.

6. PLOS authors have the option to publish the peer review history of their article (what does this mean?). If published, this will include your full peer review and any attached files.

Reviewer #1: No

Reviewer #2: No

---

## [Author Response · Author response to Decision Letter 1]

1 Apr 2026

Response to reviewers: Manuscript # PONE-D-25-56497) titled “Effectiveness of a home-based exercise program for improving upper limb function in community-dwelling older people: a pragmatic randomized controlled trial

Thank you for the opportunity to respond to the comments from the reviewers, which are listed point by point below in italics, followed by our responses. The page number and lines refer to the marked-up copy of the manuscript.

Reviewer #1: GENERAL COMMENTS

Thank you for the opportunity to review manuscript # PONE-D-25-56497) titled “Effectiveness of a home-based exercise program for improving upper limb function in community-dwelling older people: a pragmatic randomized controlled trial”. Overall, I think this is an important and well-done study and I enjoyed reading it. I am especially encouraged that the authors submitted this manuscript despite what were essentially null results. Why? Two reasons – First, the quality of any manuscript should be driven by the quality of the methods and not the results. Second, knowing what might not work, as well as what does work, are both important.

SPECIFIC COMMENTS

Comment 1

*Page 3, lines 53 and 54 (Abstract) – Since you say here in your Conclusions that people “can successfully learn and adhere to a home-based exercise...” I would suggest that you briefly report your results for adherence somewhere in the Results section of your abstract.

Response: We have addressed this in the Abstract. Added text: Participants performed the exercises twice per week, averaging 104 exercise sessions (SD 69, median 117, range 0-371) over the 12-month intervention period (page 3, lines 51-53).

Comment 2

*Page 4 (Introduction) – I think this is a nice introduction. I only have one suggestion. Somewhere in your Introduction, briefly describe, with appropriate citations, the economic costs associated with upper limb dysfunction. This would provide additional justification for your study.

Response: We have addressed this in the Introduction. Added text: The economic costs associated with shoulder pain include direct and indirect healthcare costs, work productivity, and personal and domestic support costs (Marks 2019). These costs were calculated to be between $13885 and $22378 per year for patients on an orthopaedic waiting list at an Australian public hospital in 2013/14 (Marks 2019) (page 4, lines 69-73).

Comment 3

*Pages 8 through 14 (Materials and Methods) – In either a separate section at the end of your Methods or throughout your Methods section, please describe any post hoc changes to your a priori protocol, including the reasons for those changes.

Response: No changes to the protocol were made.

Comment 4

*Page 9, lines 195 and 196 (Materials and Methods, Study Design) – Since you followed the CONSORT guidelines, and in addition to reference #39 (CONSORT 2010 Statement), it would be better to also include the CONSORT guidelines extension statement for pragmatic trials. Here’s the reference: Zwarenstein M, Treweek S, Gagnier JJ, et al. Improving the reporting of pragmatic trials: an extension of the CONSORT statement. BMJ. 2008;337:a2390. Doi: 10.1136/bmj.a2390. This may also be helpful to you in the reporting of your entire study.

Response: Thank you. The additional reference (Zwarenstein) has been added and updated CONSORT 2025 statement (page 10, line 201).

Comment 5

*Pages 10 through 12, lines 217 through 252 (Materials and Methods, Intervention) – Please specify the number of sets of each exercise that participants performed at each session. I’m assuming one but am not sure. Assuming this is accurate, maybe on lines 219 and 220 something like …”consisted of one set of eight exercises,… versus “…”consisted of a set of eight exercises,…” Also, on lines 230 and 231 could you please describe the “strategies to progressively increase the difficulty of the exercises as they advanced through the program.” I think some readers, including myself, would be especially interested in this. Finally, on lines 242 and 243, I’m encouraged to see you mention the behavioral intervention that you used.

Response: Thank you, the sentence has been updated to read ‘The UL program consisted of one set of eight exercises’ (page 11, line 223).

Description of the strategies used to increase the difficulty of the exercise have been added. Text reads: Participants were taught strategies to progressively increase the difficulty of the exercises as they advanced through the program, such as using a heavier weight or thicker resistance band, increasing the number of sets and/or repetitions of each exercise, increasing the range of motion of some exercises, slowing down the eccentric phase, adding a pause at the point of maximum exertion (page 11, lines 235-238.

Comment 6

*Pages 12 and 13 (Materials and Methods, Outcome measures, Primary outcome, Secondary outcomes) – In my opinion, a nice description of your primary and secondary outcomes.

*Page 14 (Materials and Methods, Data analysis) – Overall, a nice data analysis section. However, while you mention clinical relevance back in your abstract as well as in your Results, I do not see any information regarding what you considered to be a minimal clinically important difference (MCID) for your primary and secondary outcomes. I do however see data regarding a statistically significant difference for your primary and secondary physical outcomes, i.e., “…10% between-group difference in the DASH total score and the secondary physical outcomes, with a 15% loss to follow up (38).” Please elaborate.

Response:

Information on the minimal clinically important difference for the DASH (from a 2024 systematic review and meta-analysis) has been added to the manuscript. Added text: A change of approximately 12-14 points on the DASH is generally considered the minimal clinically important difference, representing the smallest change perceived as beneficial by people with upper limb disorders (Galardini et al 2024) (page 13, lines 274-276).

Comment 7

Also, please describe exactly what linear regression models you used with appropriate citations where necessary. Furthermore, please tell the reader if any adjustment was made for the multiple statistical tests you conducted and how this was done. If no adjustments were made, then this should also be stated. Finally, on line 303, please tell the reader if your tests for statistical significance were one-tailed or two-tailed.

Response: Thank you for your comment. We have revised this section to clarify and add appropriate citation. Vickers and Altman (2001) reference has been added to justify the linear regression model (page 15, line 306).

Regarding multiple statistical testing, no formal adjustment was applied. The outcomes analysed were prespecified secondary outcomes, and the analyses were considered exploratory. This has now been stated in the manuscript. Text added: No formal adjustment for multiple comparisons was undertaken (page 15, line 313).

We have also clarified that statistical tests were two-tailed with statistical significance set at p < 0.05. Text added: All statistical tests were two-tailed with statistical significance set at p <0.05 (page 15, line 312).

Comment 8

*Page 15 (Table 2) – It would be nice to see 95% confidence intervals for this data.

Response: Thank you for this suggestion. However, we have not included confidence intervals for the baseline characteristics as these data are intended to be descriptive rather than inferential. The 2025 CONSORT Statement recommends presenting baseline demographic and clinical characteristics for each group in a table using descriptive statistics (e.g., mean and standard deviation for continuous variables, or counts and percentages for categorical variables). The CONSORT explanation and elaboration also notes that standard errors and confidence intervals are inferential statistics and are therefore not appropriate for describing variability in baseline characteristics, and that statistical testing of baseline differences between randomised groups should not be performed.

Because participants are randomly allocated, any baseline differences are expected to occur by chance, and formal statistical inference for baseline comparisons is therefore considered unnecessary and potentially misleading. For this reason, we have retained descriptive summaries only for the baseline characteristics table, consistent with CONSORT reporting recommendations (Hopewell 2025 – CONSORT 2025).

Comment 9

*Page 17 (Results, Effect of intervention, Secondary outcomes) – Please tell the reader if the differences you observed were considered to be clinically important.

Response: This text has been added: However these differences were unlikely to be clinically important. (page 17, line 364)

Comment 10

*Page 18 (Table 3) – In addition to absolute difference, please also report relative, i.e., percent, changes.

Response: Thank you for your suggestion. However, this is not standard or recommended practice either, particularly since in our study most outcomes did not change. (Vickers 2001) and is not required by CONSORT (Hopewell 2025). We feel that Table 3 is already large, and this would not add to the manuscript.

Comment 11

*Page 20 (Results, Adherence with the program) – Overall, a very nice description of adherence. I just have one suggestion. In the first paragraph, suggest that you report 95% confidence intervals for your percent adherence data.

Response: Thank you for this helpful suggestion. We agree that confidence intervals are valuable for conveying the precision of key study estimates. However, the attendance values reported in this paragraph describe observed participation in the intervention (i.e. program adherence) rather than an inferential estimate or treatment effect. Consistent with reporting guidance for randomised trials, including the CONSORT Statement and intervention reporting frameworks such as the TIDieR Checklist, intervention adherence data are typically reported descriptively using counts and percentages. For this reason, we have retained the descriptive presentation of attendance data without confidence intervals.

Comment 12

*Page 21 (Acceptability of the intervention) – Overall, a very nice description here. Along those lines, it does not appear that kinesiophobia is a major issue for these folks.

*Pages 21 through 25 (Discussion) – Overall, a nice Discussion. However, for ease of reading, I would suggest that you partition your Discussion into the following subsections and address: (1) Overall findings, (2) Implications for research, (3) Implications for practice, (4) Implications for policy, if any, and (5) Strengths and potential limitations.

Response: Additional headings have been added: Overall findings and Strengths (pages 22-25).

Comment 13

Also, I think it is important to point out somewhere that “absence of evidence” is not “evidence of absence”.

Response: Thank you for this suggestion. We agree that non-significant findings should be interpreted cautiously. We have revised the Discussion to clarify that the absence of statistically significant differences between groups should not be interpreted as evidence that the intervention has no effect.

This has been incorporated. Text added: Although no statistically significant between-group differences were observed, these findings should not be interpreted as evidence that the intervention has no effect, rather, the results indicate that this study did not detect a statistically significant difference between groups (page 24, line 485).

Reviewer #2

Comment 14

Introduction

- When referring to articles consider adding the date ex: Kim et al., (XXXX)

Response: This has been updated in the manuscript.

Comment 15

- I think the introduction can be made more concise. Consider condensing the concepts of shoulder dysfunction as well as condense your rationale for exercise interventions in the shoulder. Refer to systematic reviews and meta-analyses were appropriate to help summarize the literature more concisely.

Response: Thank you for your suggestion, however there is such limited literature in this area, specifically related to older adults that we felt it was relevant to highlight the few individual studies that have focused on older adults.

Comment 16

- Maybe highlight the literature around the DASH to help justify it’s use as your primary outcome

Response: The DASH is a validated patient-reported outcome measure, further detail and references on the DASH has been provided in Methods – Outcome measures, stating that the DASH is a validated tool (page 13, line 265) and information related to reliability, sensitivity and responsiveness to change, with additional references (page 13, lines 273-276).

Comment 17

Methods:

- I wonder if the Otago program was challenging enough for this demographic for the LL group, maybe biasing the upper extremity group

- Include a sample size calculation

- Lines ~217–233, 388–403

Response: This is in data analysis section of the manuscript (page 15, lines 315-320).

Text: Sample size calculations suggested that 576 participants would be sufficient to detect a 10% between-group difference in the DASH total score and the secondary physical outcomes, with a 15% loss to follow up (39). The sample size calculation used the nbpower user written command in Stata (Stata Statistical Software: Release 15. College Station, TX: StataCorp LLC).

Comment 18

• The intervention relies on limited supervision (three sessions over 12 months).

• Average adherence (~2 sessions/week) falls below the prescribed dose.

• Exercises were performed seated, with modest resistance

• Clarify whether the intervention was designed as maintenance, prevention, or strength-building, and explicitly discuss whether the dose and progression were sufficient to elicit physiological adaptation in older adults.

- Strength and ROM were assessed only to 6 months, whereas DASH extended to 12 months.

Response: The intervention was designed to improve strength and prevent shoulder disability in untrained older adults. Participants were provided with strategies to progress their exercises, however no further instruction was provided beyond the initial 3 sessions.

Description of strategies used to increase the difficulty of the exercise are added. These approaches are consistent with established principles of progressive resistance training aimed at promoting physiological adaptation.

Text added: Participants were taught strategies to progressively increase the difficulty of the exercises as they advanced through the program, such as using a heavier weight or thicker resistance band, increasing the number of sets and/or repetitions of each exercise, increasing the range of motion of some exercises, slowing down the eccentric phase, adding a pause at the point of maximum exertion (page 11, lines 235-238).

Additional text has also been added to the discussion.

Text added: The intervention was intentionally designed as a pragmatic trial with minimal participant contact to reflect a scalable community-based program. While participants were provided with guidance on progression, the relatively low level of supervision may have limited the extent to which some individuals progressed exercise intensity sufficiently to elicit measurable physiological adaptations. It is therefore possible that a higher training dose, more structured progression, or greater supervision may be required to optimise strength and functional improvements in older adults (page 26, line 530-537).

Comment 19

Results:

- Lines ~349–364

• Results focus on statistical significance without adequate discussion of clinical relevance, particularly given low baseline impairment.

Response: This text has been added to the results: However, these differences were unlikely to be clinically important (page 17, line 364).

Comment 20

- Lines ~375–386

• The interaction with prior falls is interesting but underpowered and exploratory.

• Interpretation risks overstating importance.

Response: The following text has been added to the results – Sub-group analyses for the primary outcome. Text added: These subgroup findings should be interpreted with caution as the trial was not powered to detect su

---

## [Decision Letter · Decision Letter 1]

19 Apr 2026

Effectiveness of a home-based exercise program for improving upper limb function in community-dwelling older people: a pragmatic randomized controlled trial

PONE-D-25-56497R1

Dear Dr. Bates,

We’re pleased to inform you that your manuscript has been judged scientifically suitable for publication and will be formally accepted for publication once it meets all outstanding technical requirements.

Kind regards,

Gurkan Gunaydin

Academic Editor

PLOS One

Additional Editor Comments (optional):

Reviewers' comments:

Reviewer's Responses to Questions

**Comments to the Author**

1. If the authors have adequately addressed your comments raised in a previous round of review and you feel that this manuscript is now acceptable for publication, you may indicate that here to bypass the “Comments to the Author” section, enter your conflict of interest statement in the “Confidential to Editor” section, and submit your "Accept" recommendation.

Reviewer #1: (No Response)

Reviewer #2: All comments have been addressed

2. Is the manuscript technically sound, and do the data support the conclusions?

Reviewer #1: Yes

Reviewer #2: Yes

3. Has the statistical analysis been performed appropriately and rigorously? 

Reviewer #1: Yes

Reviewer #2: Yes

4. Have the authors made all data underlying the findings in their manuscript fully available?

Reviewer #1: Yes

Reviewer #2: No

5. Is the manuscript presented in an intelligible fashion and written in standard English?

Reviewer #1: Yes

Reviewer #2: Yes

6. Review Comments to the Author

Reviewer #1: GENERAL COMMENTS

Thank you for the opportunity to review revised manuscript # PONE-D-25-56497R1 titled “Effectiveness of a home-based exercise program for improving upper limb function in community-dwelling older people: a pragmatic randomized controlled trial”. In my opinion, the authors have done an excellent job in responding to my previous comments and suggestions, including an appropriate response to not adhering to several of my previously suggested changes. I only have one remaining comment, detailed below.

SPECIFIC COMMENT

*Pages 9 through 15, lines 184 and 328 (Materials and Methods) – Thank you for the information that there were no changes to your a priori protocol. However, I would suggest that you also insert this information at beginning or end of your Methods so that the reader is aware of such.

END OF REVIEW

Reviewer #2: Thank you for the opportunity to re-review the manuscript. The manuscript is well written and all my comments were addressed. I have no further comments.

7. PLOS authors have the option to publish the peer review history of their article (what does this mean?). If published, this will include your full peer review and any attached files.

Reviewer #1: No

Reviewer #2: **Yes:**Christina Ziebart

---

## [Editor Report · Acceptance letter]

PONE-D-25-56497R1

PLOS One

Dear Dr. Bates,

I'm pleased to inform you that your manuscript has been deemed suitable for publication in PLOS One. Congratulations! Your manuscript is now being handed over to our production team.

Kind regards,

on behalf of

Assoc. Prof. Gurkan Gunaydin

Academic Editor

PLOS One